# Wine Intake in the Framework of a Mediterranean Diet and Chronic Non-Communicable Diseases: A Short Literature Review of the Last 5 Years

**DOI:** 10.3390/molecules25215045

**Published:** 2020-10-30

**Authors:** Simona Minzer, Ramon Estruch, Rosa Casas

**Affiliations:** 1El Pino Hospital, Avenida Padre Hurtado, San Bernardo, 13560 Santiago de Chile, Chile; simona.minzer@gmail.com; 2Department of Internal Medicine, Hospital Clinic, Institut d’Investigació Biomèdica August Pi i Sunyer (IDIBAPS), University of Barcelona, Villarroel, 170, 08036 Barcelona, Spain; restruch@clinic.cat; 3Center for Biomedical Research Network (CIBER) 06/03, Fisiopatología de la Obesidad y la Nutrición, Instituto de Salud Carlos III, 28029 Madrid, Spain

**Keywords:** Mediterranean diet, wine intake, cardiovascular disease, cancer, dementia

## Abstract

Dietary habits are a determining factor of the higher incidence and prevalence of chronic non-communicable diseases (NCDs). In the aim to find a possible preventive and intervention strategy, the Mediterranean diet (MedDiet) has been proposed as an effective approach. Within the MedDiet, moderate wine consumption with meals is a positive item in the MedDiet score; however, recent studies have reported a dose-response association between alcohol consumption and higher risk of a large number of NCDs. This review aimed to evaluate the association between NCDs and wine consumption in the framework of the MedDiet, with a simple review of 22 studies of the highest-level literature published over the last five years. We found that the information regarding the effects of wine in different health outcomes has not varied widely over the past five years, finding inconclusive results among the studies evaluated. Most of the literature agrees that light to moderate wine intake seems to have beneficial effects to some extent in NCDs, such as hypertension, cancer, dyslipidemia and dementia, but no definitive recommendations can be made on a specific dose intake that can benefit most diseases.

## 1. Introduction

Lifestyle, including dietary habits, is a determining factor of the high incidence and prevalence of chronic non-communicable diseases (NCDs), such as cardiovascular disease (CVD), diabetes and dementia [1]. Several studies have shown that non-smokers, the practice of physical activity, having an adequate body mass index (BMI) and a healthy diet pattern (rich in vegetables, fruits, legumes and fish, and limited in red meat), with moderate alcohol consumption, are factors associated with a lower risk of mortality by all causes and CVD (≥65%) compared to subjects who have an unhealthy lifestyle [2,3,4]. Moreover, the adoption of a healthy lifestyle is associated with a longer life expectancy free of the main NCDs (cancer, CVD and type 2 diabetes [T2DM] [5].

In this sense, the Mediterranean diet (MedDiet), one of the most widely evaluated dietary patterns throughout different clinical trials, and prospective cohort studies and has been proposed as an effective approach for CVD prevention and intervention [6,7,8]. This healthy dietary pattern is recommended by the American Heart Association and has been included in the 2015–2020 Dietary Guidelines for Americans [6,9]. The MedDiet has been associated with better long-term weight control, blood pressure (BP), lipid profile, glucose metabolism and insulin resistance, inflammation, endothelial dysfunction, the presence of arrhythmia, and gut microbiome [6,8], and a significant reduction in all-cause mortality (8–10%), and the risk of CVD (10%) and neoplasias (4%) for every two-point increment in adherence to the Mediterranean diet pattern (MDP) [10,11].

The MedDiet is characterized by a high consumption of plant foods, that mainly comprises fruits and vegetables, whole-grain cereals and breads, nuts and seeds; locally grown, fresh and seasonal, unprocessed foods; primarily extra-virgin olive oil as a main source of healthy fat for cooking and dressing meals; moderate amounts of dairy products; small amounts of red meat, moderate amounts of fish; and light-to-moderate intake of red wine often during the main meals [12].

Although moderate consumption of wine with meals (≤1 and ≤2 drinks/day for women and men, respectively) is a positive item in the MedDiet score, results from recent studies have reported that there is a dose-response association between alcohol consumption and a high risk of a large number of NCDs [13,14]. Specifically, a certain association has been observed between alcohol consumption and the risk of developing different types of cancer, regardless of the amount consumed [15]. On the other hand, some authors have reported that alcohol consumption is associated with a decreased incidence of cancer because of its stilbenoid content [16,17,18,19,20,21,22].

According to the World Health Organization (WHO), the amount and pattern of alcohol consumption plays a key role in the appearance of detrimental effects on health [14]. Excessive alcohol consumption is not only linked to an increased risk of traffic accidents and deaths, child abuse, spousal violence and suicide [23] but also to the development of cancer and liver disease [24]. In 2016, the WHO reported that CVD death was directly associated with excessive alcohol consumption, accounting for 19% of deaths [14]. In addition, other determining factors on the harmful effects of alcohol on health are gender (male or female), inter-individual variability, type of alcoholic beverage consumed (fermented or distilled), amount (low, moderate, high, excessive) and duration of consumption, drinking patterns (occasional, daily, compulsive), as well as socioeconomic factors [23,25,26,27]. Furthermore, a systematic review and meta-analysis analyzing 2865 participants performed by Roerecke et al. [28] reported that participants who alcohol intake >6 drinks/day leads to reduction of BP in a dose-dependent manner with an apparent threshold effect at 2 drinks/day. Drinkers (≤2 drinks/day) did not report significant reduction in BP after reducing their consumption near abstinence. The same authors in a new systematic review and meta-analysis of cohort studies [29], including 361,254 participants and 90,160 incident cases of hypertension, concluded that the risk of hypertension was increased for any alcohol consumption in men, in contrast to women who showed no greater risk with a daily consumption of 1 to 2 drinks or higher. According to results reported by Wood et al. [30], in high-income countries, among current alcohol drinkers, 100 g of alcohol per week, showed a lower risk threshold for all-cause mortality. In addition, exploratory analysis showed among alcoholic beverages that beer or spirits drinkers and binge drinkers have a higher risk of all-cause mortality. Nonetheless, wine, a fermented drink, consumed during meals and in the context of a MDP, has shown favorable effects on the prevention of CVD. These cardio-protective mechanisms involve not only an increase of high-density lipoprotein cholesterol (HDL-C) levels and the regulation of blood lipids but also improvement in glucose metabolism and endothelial function, decreasing inflammation and platelet aggregation as well as exerting antioxidant effects [31,32,33]. Moreover, wine consumption in the context of a healthy diet, such as the MedDiet, has shown positive effects against cancer risk. Results of a meta-analysis with an overall population of 2,130,753 subjects [34] reported that moderate wine consumption in the framework of a MDP has a higher protective effect (relative risk (RR) 0.89; 95% CI: 0.85–0.93) compared to the rest of its components (fruit, vegetable and whole grain intake).

Wine is an alchemy with unique properties, having a rich and original composition in terms of known polyphenols and antioxidants [35,36,37,38,39]. The alcohol content varies among different types of wines, being 14% for red wine and 11% for white wine, a much lower content than spirits (approximately 35%) [40,41,42,43]. In addition to ethanol, the polyphenol content in wine can provide a greater protective effect on health [44,45,46]. While red wine has a high concentration of bioactive compounds, the content in white wine is lower and is practically negligible in distilled beverages (liquors and spirits) [40]. Resveratrol, anthocyanins (ANC), catechins, and tannins (proanthocyanidins and ellagitannins) are the main polyphenols in wine [35,47]. Since alcoholic beverages present a different molecular composition, different health effects are to be expected. Besides wine, other bioactive components from the MedDiet, such as polyphenolic compounds and phytosterols in olive oil (hydroxytyrosol, tyrosol, oleocanthal), and nuts, fruits and vegetables (flavonoids, mainly), can also contribute to increasing this cardioprotective effect through different synergic mechanisms [47].

### 1.1. Moderate Alcohol Consumption

According to the WHO and the US National Institute on Alcohol Abuse and Alcoholism, the measure most frequently used in studies is a standard drink, which is defined as the amount of alcohol an average adult can metabolize in 1 h [40,48,49]. While in the United Kingdom and Iceland a standard drink is defined as an alcoholic beverage that contains 8 g of pure alcohol (10 mL), other countries such as Australia, France, the Netherlands or Spain consider a standard drink as 10 g of pure alcohol (12.5 mL), and the United States considers it as 14 g of pure alcohol (17.5 mL) [50]. In lay terms, these amounts can be translated into 330 mL of beer (~5% ethanol), 125 mL of table wine (~12% ethanol), or 40 mL of distilled spirits or liquor (~40% ethanol) [50,51]. Moreover, the alcohol content in wine can vary from 5% to 15% according to the different formulations [40].

The definition of moderate alcohol consumption may vary depending the country referred, as may the amount of alcohol content in an alcoholic drink. The American Dietary Guidelines Advisory Committee [52] consider a moderate alcohol intake by adults of legal drinking age as a daily amount consumed ≤10 g of ethanol (≤1 drink) for women and ≤20 g of ethanol (≤2 drinks) for men. However, other guidelines consider a low-risk pattern as a daily consumption of 10 g up to 42 g of alcohol (1 to 3 drinks) for women or 10 g to 56 g for men (1 to 4 drinks).

Furthermore, besides the amount of alcohol consumed, other aspects such as age (young people engage in more heavy drinking episodes than older individuals), sex (women are more sensitive to the toxic effects of alcohol), ethnicity, genetics, type of alcoholic beverage consumed (wine, beer, distilled), drinking frequency (heavy or binge, occasional, daily, weekend, etc.) and socioeconomic level, account for the inter-individual variability of the adverse effects of alcohol on health [23,25,27,51,53].

### 1.2. Wine Polyphenols in Human Health

Wine matrix composition is complex and mainly constituted by water (86%), ethanol (8–15%), glycerol and polysaccharides or other trace elements (1%), different types of acids (0.5%), and a volatile fraction (0.5%) [54]. The wine matrix contains several hundred compounds that are found in very low concentrations, and they have been found to play an important role in the evolution and quality of wine, as well as in the protection against NCDs [55].

Among the minority compounds, wine contains a large variety of phenolic compounds (range from 2000 to 6000 mg/L in red wines), also called polyphenols, which are responsible for special organoleptic features of wine (color, flavor, smell) [52,55]. Currently, there is a great interest in the volatile fraction of wine as these compounds are closely related to beverage flavor [56]. These volatile organic compounds (VOC) include alcohols, esters, aldehydes, ketones, acids, terpenes, phenols, and sulfur compounds in a great variety of concentrations that are secondary metabolites produced in grape plants as a defense mechanism [57]. The perception of aroma and flavor are a result of a complex interaction between the volatile (which includes flavor and aroma compounds) and nonvolatile (ethanol, polyphenolic compounds, proteins, and carbohydrates) fractions in wine [57,58]. It is estimated that wine contains more than a thousand volatile compounds, whose concentrations range between mg/L to ng/L [58]. Wine flavor is obtained by varietal aroma, grape variety, pre-fermentative (during alcoholic and malolactic fermentations) and post-fermentative aroma (during conservation and aging of wine) [54,59]. Wine aroma is quantitatively produced by higher alcohols, acids and esters, which are important for the sensory properties as quality of wine, intrinsic factors that influence consumer acceptance [54,60]. While the higher amount of alcohol is directly associated with wine quality (higher amounts, less wine quality), the amount of esters (generally ≤100 mg/L) is associated with wine odor (higher amounts, strong odor) [54,61]. Wine oligosaccharides (complex carbohydrate molecules) have been associated with significant physicochemical properties beneficial to consumers’ health. So, some oligosaccharides such as arabinoxylan-, fructo-, gluco-, galacto-, isomalto-, mannan-, xylo-, soyo-oligosaccharides and others can be fermented, exerting benefits on the intestinal microbiota (prebiotic effect) [62,63]. In addition, it has been suggested pectin-derived acidic oligosaccharides and arabinoxylooligosaccharides may have anti-cancer [64] and antioxidant effects [65,66]. Similarly, it has been suggested that polysaccharides could also have a significant antioxidant effect in wine [63]. In addition, some of these VOCs are sesquiterpenes and monoterpenes, which have been shown to have potential health benefits, such as decreased risk of chronic diseases. These compounds have been associated with anti-inflammatory, antioxidant, anti-carcinogenic and anti-bacterial properties [67,68,69,70], contributing to wine’s health effects.

During the red winemaking process the contact with grape skins and seeds is longer, and therefore, red wines tend to have a higher polyphenol content (six-fold greater) than white wines. It is estimated that the polyphenolic compound content in red wine varies from 1800 to 3000 mg/L [71]. Polyphenols, especially flavonoids such as flavonols (quercetin and myricetin), flavanols (catechin and epicatechin) and ANC, and non-flavonoids, which include phenolic acids (hydroxybenzoic acids and hydroxycinnamic acids) and stilbenes (trans-resveratrol), have been related to beneficial effects on human health due to their protective properties, antioxidant activity and capacity to delete reactive oxygen species (ROS) caused by exercise, food metabolism and environmental factors, such as exposure to air pollutants. These free radicals can lead to aging, cardiovascular and neurodegenerative diseases and even cancer by the reduction of cell proliferation, which can be used for potential cancer therapy [36,55,71].

Even though the alcoholic fraction of wine (ethanol) has been associated with pro-oxidant effects, the phenolic content (polyphenols) seems to counteract the potential pro-oxidant effect of ethanol [38]. The antioxidant capacity of wines is only associated to its phenolic content or the action of a single phenolic compound but within the total polyphenol content (synergistic antioxidant effect) [55].

To date, the main biological effects attributed to phenolic acids (gallic acid or caffeic acid) are their antioxidant, anti-mutagenic, anti-proliferative and antimicrobial properties [36]. In addition, in vitro studies have reported vasodilator activity of phenolic acids [36,71]. However, Mudnic et al. [72] found a negative correlation between antioxidant activity and vasodilatory capacity after testing nine different phenolic acids. Moreover, caffeic acid has been associated with neuroprotective activity [73] and inhibition of peroxynitrite-induced neuronal injury, and ferulic acid is considered to have antidiabetic properties [74] because of its capacity to reduce blood glucose levels through increasing plasma insulin concentrations.

Flavonols (quercetin, mainly) found in red wine have an approximate concentration of 50 mg/L [75]. The beneficial effects of dietary flavonol on human health have been related to the inhibition of low-density lipoprotein cholesterol (LDL-C) oxidation and a reduction of oxidative stress through decreases in BP, which are primary risk factors for the development of atheroma plaque [71,76]. This flavonol is able to reduce oxidative stress through an upregulation of nitric oxide synthase (NOS) expression, as well as an activation and modulation of antioxidant mechanisms. In addition, quercetin has been associated with decreased inflammation, reduction of the expression of Toll-like receptors (TLR2 and TLR4) by the inhibition of nuclear factor kappa-B’s (NF-κB) translocation to the nucleus. Quercetins can inhibit cell proliferation, which leads to an attenuation of the progression of cancer [55]. Besides their anti-hypertensive and anti-atherogenic effects, flavonols have been inversely related to aging, obesity and the occurrence of neurodegenerative diseases, CVD and several specific types of cancer such as breast, pancreatic, uterine, prostate or urinary tract cancer, among others [36,55,77,78,79,80,81].

It has also been demonstrated that ANC are beneficial to human health. Their concentration in red wine is approximately 500 mg/L [82]. ANC are strong antioxidants and have the capacity of inhibiting cancer cell growth, inflammation, neuro-inflammation and oxidative stress, as well as preventing obesity [83,84,85,86,87]. Finally, stilbenes are bioactive compounds with concentrations in red wine of approximately 20 mg/L [88]. The main function of stilbenes in plants is to protect them against pathogens and fungi, and therefore, they present a strong antifungal and antimicrobial capacity [89]. The most important stilbene is trans-resveratrol, as it presents multiple relevant pharmacological effects on health. These compounds (resveratrol, mainly) present anti-inflammatory, anti-oxidative and anti-aggregatory effects, as well as a high capacity for modulating lipoproteins and inhibit the initiation, promotion and progression of tumors. Therefore, their biological activity has frequently been related to atherosclerosis, cancer, CVD or neurodegenerative diseases (e.g., Alzheimer’s disease) [71,90,91,92,93]. In addition, resveratrol is associated with a lower risk of coronary heart disease (CHD) and myocardial infarction [94], and several clinical studies and meta-analyses have found a significant reduction of systolic BP (SBP) with resveratrol intake. Therefore stilbenes might have an protective role against hypertension, as well as diabetes and diabetes-related complications [95,96,97,98,99].

There is noteworthy information suggesting that the potential benefits of wine intake on NCDs such as dyslipidemia, hypertension, MetS, CVD and T2DM, are dependent on the bioavailability of polyphenols [55]. Phenolic compounds’ bioavailability can be affected by different factors such as environmental, dietary factors (fibers and fats that help or reduce absorption), possible interactions with others compounds of similar mechanisms of absorption. Moreover, thermal treatments, storage, cooking techniques, food matrix, chemical structure, amount of polyphenols in food could contribute to their bioavility. Therefore, others intrinsic factors such as age, gender and genetic differences, enzyme activity, transporters, intestinal microflora, health status, among others may influence [36,100,101]. So not all polyphenols are absorbed with equal efficacy, only between 5 to 10% of the total polyphenol intake may be directly absorbed in the small intestine [102,103].

This review aimed to evaluate the association between NCD (hypertension, T2DM, dyslipidemia, cancer and dementia) and wine consumption within the framework of a MDP and its underlying mechanisms of protection, with a simple review of the highest-level literature (randomized control trials (RCT) and meta-analyses) published in the last five years, evaluating humans, adults (>18 years), addressing wine intake or specifically red wine polyphenols. The bibliographic search was performed through PubMed, ScienceDirect, and Google Scholar from June 2020 to August 2020.

## 2. Results

A total of 22 studies were selected for the final evaluation; seven studies evaluating hypertension as an outcome, eight studies on T2DM, four studies on dyslipidemia, four studies on cancer and one study on dementia. Some of the studies evaluated two or more outcomes, thus overlapping in the results. The study characteristics are summarized in Table 1. 

### 2.1. Hypertension

Alcohol intake has been associated with BP levels in a J-shaped form, in which low to moderate intake contributes to lower BP levels, and higher intakes increase these levels [40,49]. A RCT by Gepner et al. evaluated the effect of moderate red wine intake on BP in individuals with T2DM who abstained from alcohol intake. After a 6-month intervention, reductions in BP were observed in the red wine group at midnight (SBP −10.6 mmHg, 95% confidence interval (CI) −14.1 to −0.6; *p* = 0.03 and diastolic (DBP) −7.7 mmHg, 95% CI −11.8 to 0.9; *p* = 0.076) and at 7 to 9 am (SBP −6.2 mmHg, 95% CI −17.3 to −0.8; *p* = 0.014), but no long-term effects in the mean 24-h BP were found [104]. Other studies using ambulatory BP monitoring have described a biphasic BP pattern after alcohol intake, showing lower levels after acute ingestion but higher levels after 13–23 h [123]. In 2016, Mori et al. published a RCT that evaluated the effects of red wine (24–31g of alcohol/day) over four weeks on BP levels in 24 adults with T2DM. The authors described that red wine significantly increased awake SBP and DBP compared to water (2.5 ± 1.2 mmHg vs 1.9 ± 0.7 mmHg; *p* = 0.033 and *p* = 0.008 respectively), but decreased DBP during sleep (2.0 ± 0.8 mmHg; *p* = 0.016), resulting in a non-significant overall effect on the mean 24-h SBP and DBP [105].

Within the MDP, high intakes of green leafy vegetables are recommended. Dietary nitrate (NO_3_^−^) and nitrite (NO_2_^−^) intake, present in vegetables, can affect BP levels by increasing plasma nitric oxide (NO) production and help reduce BP in a dose-dependent manner [106,107]. NO can reduce vascular oxidative stress and act as a natural vasodilator. It has been described that red wine intake along with NO_3_^−^ or NO_2_^−^ promotes NO formation, attributing this phenomenon to its polyphenol content [106,124]. In 2018, McDonagh et al. published a RCT evaluating the response to a NO_3_^−^ rich meal associated with red wine (175 mL), compared to vodka or water intake, in 12 healthy normotensive males. Results showed that compared with the controls SBP decreased at 2 h after consumption of red wine (111 ± 7 mmHg vs. 116 ± 6 mmHg; *p* < 0.05) and at 5 h (115 ± 8 mmHg versus 119 ± 7 mmHg), and DBP was also significantly reduced after consumption of red wine (1h: −3 mmHg, 2h: −4 mmHg) compared with baseline and also compared with controls (1h: 59 ± 5; 2h 58 ± 4 mmHg vs. 1h: 62 ± 5, 2h: 61 ± 6 mmHg; *p* < 0.05). Moreover, the mean arterial pressure was reduced 2 h after intake of the NO_3_^−^ rich meal alongside red wine (76 ± 4 mmHg) compared with baseline (80 ± 6 mmHg) and controls (80 ± 6 mmHg; *p* < 0.05) and also at 3 h (77 ± 6 mmHg) compared with controls (81 ± 7 mmHg). Even though vodka was also effective in reducing BP, the magnitude of the SBP reduction was consistently higher after red wine intake [106]. Another crossover trial by Roth et al. evaluated the effects of short-term white wine and gin intake (21 days) on BP and plasma NO in 41 adult men (55 to 80 years) with cardiovascular risk factors. All participants presented low to moderate alcohol consumption, quantified around 30 g ethanol per day. After the intervention, white wine intake showed a significant mean reduction in SBP (−4.91 mmHg, 95% CI −9.41 to −0.42; *p* = 0.033) and DBP (−2.90, 95% CI −5.50 to −0.29; *p* = 0.030), and this reduction was also significant when compared to gin intake (*p* < 0.040). Moreover, plasma NO concentrations significantly increased after white wine intake (27.86, 95% CI −6.86 to 62.59; *p* = 0.013), but no differences were observed between the two groups. The authors suggested that the hypotensive effects of white wine evaluated in this study could be attributed to non-alcohol compounds found in this beverage [107].

It has been described that the phenolic content of wine contributes to its BP lowering effect. In vitro studies in human endothelial cells have described that wine polyphenols inhibit nicotinamide adenine dinucleotide phosphate (NADPH) oxidase activity and increase calcium intracellular concentrations and NO synthesis, contributing to the vasorelaxant effect attributed to wine [125,126]. Red wine is a rich source of ANC, an important type of polyphenols. Multiple cardio-metabolic components can be altered in response to ANC food products, and their intake has been associated with a lower risk of CVD [108]. Studies evaluating the effects of ANC and other polyphenols on cardio-metabolic risk factors (i.e., serum lipids, blood glucose levels, insulin resistance, hypertension, among others) show inconsistent results, depending on various factors, such as the health status of the participants, sources of bioactive compounds, type and dose of polyphenols supplied, their bioavailability, and host characteristics. A meta-analysis by García-Contesa et al. studied the association between various food sources of ANC, including red wine, with different biomarkers of cardio-metabolic risk. The study included 128 human RCTs, with a total of 5538 participants. The authors found that the intake of specific sources of ANC significantly reduced SBP (−3.31; *p* = 0.014) and DBP (1.50; *p* = 0.002), specifically red wine; nonetheless red wine did not reduce total cholesterol or glycated hemoglobin (HbA1c) [108].

Another important polyphenol present in wine is resveratrol, which has been widely studied. A meta-analysis published by Weaver et al. evaluated the effects of red wine polyphenols, especially resveratrol, on vascular health. Supplementation with red wine polyphenols significantly decreased SBP (−2.62 mmHg, 95% CI −4.81 to −0.44; *p* = 0.010), but group analysis showed this effect only in at-risk populations (−3.2 mmHg, 95% CI −5.7 to −0.8; *p* = 0.010) and not in healthy cohorts (0.7 mmHg, 95% CI −2.5 to 3.8; *p* = 0.673). When analyzing resveratrol-only studies, this significant mean difference was maintained (−3.7 mmHg, 95% CI −7.3 to −0.0; *p* = 0.047) but was not seen in the non-resveratrol group. No significant effects were found for DBP (−1.0 mmHg, 95% CI −2.2 to 0.3; *p* = 0.139). The authors suggested that the non-significant effect on DBP could be attributed to the small changes seen in DBP in clinical hypertension [109]. Nonetheless, these findings are consistent with previous studies on resveratrol [95]. Moreover, Ye et al. published a meta-analysis of 9 randomized intervention studies evaluating the effects of wine intake on BP, glucose parameters and the lipid profile of T2DM patients. The results showed no significant differences in SBP (weighted mean difference (WMD) 0.12, 95% CI −0.05 to 0.28; *p* = 0.17)), yet a reduction was seen in DBP levels (WMD 0.10, 95% CI: 0.01 to −0.20; *p* = 0.03). These findings were attributed to the fact that the results were pooled as an average, rather than evaluated at individual hours [110].

These studies published over the last five years indicate that, overall, wine intake helps reduce BP. Even though this effect could be different depending on the time of day and time after intake, the general conclusion in most studies is that wine and its polyphenol supplementation help reduce SBP and DBP when taken in light to moderate quantities.

### 2.2. Type 2 Diabetes Mellitus

Epidemiological studies suggest that the risk of T2DM is decreased in moderate alcohol drinkers [111,112]. Alcohol, especially wine, has been associated with enhanced glycemic control [127,128]. A RCT study by Gepner et al. recruited 224 alcohol-abstaining adults 40 to 75 years of age with T2DM. The subjects were randomized to consume mineral water, white wine or red wine (150 mL) and were followed during 24 months. The authors described that both types of wine tended to improve glucose metabolism, yet only white wine significantly decreased fasting plasma glucose (−17.2 mg/dL, 95% CI −28.9 to −5.5 mg/dL; *p* = 0.004) and the Homeostatic Model Assessment for Insulin Resistance (HOMA-IR) score by 1.2 (95% CI −2.1 to −0.2; *p* = 0.019) compared with the water group. Red wine did not significantly decrease these measures. No changes were observed in HbA1c% for either type of wine. The authors suggested that the effect of wine seen on glycemic control was mainly due to alcohol [111].

It has been suggested that the glucose lowering effect of alcohol may be mediated by the incretin effect (glucose-dependent insulinotropic polypeptide (GIP) and glucagon-like peptide 1 (GLP-1)). Abraham et al. carried out an interventional study evaluating if acute red wine intake affects glycemic control during an oral glucose tolerance test and the potential involvement of incretins in the augmentation of insulin response after alcohol intake. Nine diabetic or pre-diabetic subjects consuming 263 mL of water or red wine were evaluated in a randomized crossover study. After 45 min, a higher rate of increase in glucose was observed in the wine group (0.15 ± 0.01 vs. 0.11 ± 0.01 mmol/L/min; *p* < 0.001), yet the incremental blood glucose area under the curve (iAUC) was similar in both groups (917 ± 88 vs. 904 ± 79 mmol/L/min for water and wine, respectively; *p* = 0.82). The iAUC for insulin was 50% greater after wine than after water intake (14,837 ± 4759 vs. 9885 ± 2686 μU/mL/min; *p* < 0.05), as GIP iAUC increased 25% after wine treatment (7729 ± 1548 vs. 6191 ± 1049 pmol/L/min; *p* < 0.05), with no difference in GLP-1 iAUC. The authors suggested that the higher insulin secretion after wine intake may be partially defined by an increase in GIP levels. Nonetheless, wine did not alter glucose iAUC, and therefore, glycemic control, which could be explained by the basal insulin resistance of the subjects [112].

Huang et al. carried out a meta-analysis of 13 prospective studies to evaluate the association between specific types of alcoholic beverages (wine, beer, spirits) and the risk of T2DM. Alcohol consumption was categorized into three groups of intake, low (0–10 g/day), moderate (10–20 g/day) and high (>20 g/day). The results showed that wine consumption reduced the risk of T2DM by 15% (RR 0.85, 95% CI 0.80 to 0.89). Moreover, all three categories of intake significantly decreased the risk of T2DM, showing a U-shaped relationship. For wine, all levels of consumption <80 g/day were associated with a decreased risk of T2DM, with the lowest risk at 20–30 g/day (pooled RR of moderate and high intake category 0.83, 95% CI 0.76 to 0.91). Beer consumption showed a slightly lower risk of T2DM (RR 0.96, 95% CI 0.92 to 1.0), and spirits intake did not have a significant effect on reducing the risk of diabetes (RR 0.95, 95% CI 0.89 to 1.03). The authors suggested that the greater effect of wine in the reduction of T2DM risk could be attributed to the polyphenols present in wine, especially resveratrol [113].

In the literature, moderate wine consumption has been associated with a decreased risk of metabolic syndrome and CVDs [129]. These effects have been attributed to the polyphenols present in wine, as seen in animal studies [130]. Unfortunately, these findings have been difficult to consistently observe in human studies. Woerdeman et al. performed a RCT to evaluate the effects of an 8-week supplementation with red wine polyphenols on insulin sensitivity in obese subjects (body mass index ≥30 kg/m^2^). The authors described that high dose red wine polyphenols (600 mg per day) did not alter insulin sensitivity measured by a hyperinsulinemic-euglycemic clamp, the Matsuda index and HOMA-IR, compared with placebo. The authors attributed these findings to the healthy state of the participants, the small sample size and the unknown bioavailability of the polyphenols in the study subjects [114].

The previously cited meta-analysis by García-Contesa et al. also studied the association between various food sources of ANC on HbA1c, finding that this parameter was increased (+0.97; *p* = 0.038) [108]. In a similar study by Mori et al., adults with T2DM consumed red wine, dealcoholized red wine and water during four weeks, over a three-period cross-over study. The authors described that red wine had no significant impact on home blood glucose monitoring or the HOMA-IR score, relative to dealcoholized red wine or water [105]. Moreover, in a RCT by Golan et al., 224 diabetic subjects were studied to observe if moderate wine consumption, as part of a MedDiet, would decrease their metabolic risk. After two years, the subjects assigned to white or red wine consumption showed a non-significant reduction in carotid total plaque volume (TPV) (−1.2 mm^3^, 95% CI −3.8 to 6.2; *p* = 0.6 and −1.3 mm^3^, 95% CI −3.4 to 6.0; *p* = 0.5, respectively). Among participants with the highest tertile of baseline carotid TPV, those assigned to wine consumption showed a significant regression in carotid plaque compared to baseline levels (mean −0.11; *p* = 0.04). The authors concluded that moderate wine consumption was associated with no progression in carotid plaque or de novo formation, and that a small regression could be seen in subjects with a higher baseline burden. They speculated that MedDiet counseling could have contributed to this result [115]. Finally, in a similar meta-analysis by Ye et al., the nine interventional studies evaluated showed no significant differences in fasting glucose (WMD −0.00, 95% CI −0.58 to 0.59; *p* = 1.00), fasting insulin (weighted mean difference (WMD) −0.22, 95% CI −2.09 to 1.65; *p* = 0.82) or HbA1C% (WMD −0.16, 95% CI −0.40 to 0.07; *p* = 0.17) [110].

Most studies evaluated in this review suggest that acute or long-term wine intake has little or no effects on blood glucose markers. Nonetheless, one meta-analysis did show that the overall T2DM risk decreased with wine intake [113]. More information is needed in order to provide definitive recommendations.

### 2.3. Dyslipidemia

It has been widely studied that the beneficial effects of alcohol on cardiovascular health are mainly due to its ability to raise HDL-C levels [131]. In the previously mentioned study by Gepner et al., HDL-C levels significantly increased in the red wine group (2.0 mg/dL, 95% CI 1.6 to 2.2 mg/dL; *p* < 0.001) compared to water, and both types of wine decreased triglyceride levels. The authors suggested that red wine may be superior in improving lipid variables, which may be attributed to the synergistic effect between alcohol and non-alcoholic wine compounds [104].

Red wine has also been associated with a protective effect against LDL-C for its higher antioxidant capacity compared to white wine [132]. Taborsky et al. published a RCT in 2017 in which they evaluated the long-term effects (12 months) of red and white wine intake (0.2–0.3 L per day) on biomarkers of atherosclerosis in 157 healthy subjects with mild to moderate CVD risk. They observed that HDL-C levels significantly decreased at six months in the white wine group compared to baseline (−0.14 ± 0.41; *p* = 0.005) with a significant reduction in LDL-C in both groups at 6 (−0.39 ± 0.74; *p* < 0.001 and −0.27 ± 0.68; *p* = 0.001 for white and red wine respectively) and 12 months (−0.24 ± 0.68; *p* = 0.003 and −0.24 ± 0.78; *p* = 0.013 for white and red wine respectively) compared to baseline. Total cholesterol showed a significant reduction in both groups at 6 months (−0.33 ± 0.99; *p* < 0.001 and −0.33 ± 0.82; *p* = 0.001 for white and red wine respectively), but only for red wine at the 12-month evaluation (−0.24 ± 0.82; *p* = 0.016). Triglyceride levels were not significantly altered. The authors could not confirm any clinically important differences in biomarkers of atherosclerosis between the two types of wine, and they attributed this partially to the prospective long-term design of the trial [116].

Previous studies have evidenced that red wine can modulate the inflammatory response caused by different types of meals [133]. Di Renzo et al. published a RCT in which they evaluated the effects of different types of beverages (red wine, white wine and vodka) on the oxidative status of 55 healthy subjects after a Mediterranean or high-fat meal. They found significant differences in oxidized low-density lipoprotein cholesterol (oxLDL-C) levels between the Mediterranean meal and the high-fat diet (−1.32 ± 20.43 vs. 21.29 ± 29.93%; *p* ≤ 0.05, respectively) and between the high-fat diet alone versus the high-fat diet plus red wine intake (21.29 ± 29.93 vs. −4.97 ± 33.18%; *p* ≤ 0.05, respectively). Moreover, a significant up-regulation was observed in catalase levels after red wine intake (4.04-fold change in gene expression). No other differences were observed. It was speculated that the fat profile found in the high fat meal could explain the higher levels of oxLDL-C observed, and that these could be reduced by the presence of the antioxidant molecules found in red wine. The authors also suggested that ethanol may play a role in the bioavailability of polyphenols during digestion [117].

In a similar meta-analysis by Ye et al., the intervention studies evaluated showed a significant reduction in total cholesterol following wine intake in T2DM patients (WMD −0.16, 95% CI 0.02 to 0.31; *p* = 0.03). Moreover, no significant associations were found for measures of LDL-C, HDL-C or triglycerides. The authors associated these results with the wine dose, trial duration and the study populations evaluated in the different studies [110].

The most recent studies evaluating the effect of wine on the lipid profile have shown different and inconsistent results. Even though the most accepted effect is the rising of HDL-C, one RCT and a meta-analysis failed to support this outcome. Nonetheless, oxLDL-C was lowered, contributing to the healthier cardiovascular profile attributed to wine intake within a MedDiet.

### 2.4. Cancer

It has been widely described that only 5 to 10% of all cancers can be attributed to genetic predisposition, leaving 90 to 95% of cases to be associated with lifestyle factors [134]. Among these factors, diet and more specifically, alcohol consumption have been studied as an important modifiable risk factor. Alcohol has been described to show different effects on cancer risk depending on location. A meta-analysis by Fang et al. published in 2015 evaluated the association between gastric cancer and a diversity of dietary factors, with alcohol being among them. Among the 76 prospective studies revised (n = 6,316,385 subjects and 32,758 incident gastric cancer cases), the authors found a strong effect of alcohol on gastric cancer risk (RR 1.15, 95% CI 1.01 to 1.31) when comparing the highest reported intake with the lowest, but not of wine intake (RR 1.02, 95% CI 0.77 to 1.34). The authors suggested that these findings could be associated with the protective substances found in wine [118].

Another meta-analysis by Chen et al. evaluated the dose-response association between wine intake and the risk of breast cancer. They included 26 prospective studies with a total of 18,106 breast cancer cases and found an increased risk of breast cancer (RR 1.36, 95% CI 1.20 to 1.54; *p* < 0.001) when comparing the highest versus the lowest category of wine drinking. Furthermore, a dose-response analysis showed that greater wine intake could lead to a higher risk of breast cancer, with a 0.59% of non-significant risk increase for every 1g of ethanol per day derived from wine (RR 1.0059, 95% CI 0.9670 − 1.0464; *p* = 0.6156). Moreover, the lowest risk was observed in women who consumed <40 g of wine per day (<5 g of ethanol) [119].

Vartolomei et al. published a meta-analysis in 2018 in which they studied the effect of moderate wine consumption on prostate cancer. They included 17 studies and found a pooled RR for prostate cancer risk of 0.98 (95% CI 0.92 to 1.05; *p* = 0.57). When evaluating each type of wine, the authors described that moderate white wine intake increased the risk of prostate cancer (pooled RR 1.26, 95% CI 1.10 to 1.43; *p* = 0.001) and moderate red wine consumption decreased this risk (pooled RR 0.88, 95% CI 0.78 to 0.999; *p* = 0.047). Nonetheless, it must be taken into account that this study is based in non-randomized observational studies, implying possible bias [120]. Another meta-analysis by Xu et al. evaluated the effects of wine intake on the risk of colorectal cancer in 17 observational studies (12,110 colorectal cancer cases). The risk of developing colorectal cancer due to wine consumption was RR 0.99 (95% CI 0.89 to 1.10) compared to non-drinkers. The authors did not find any associations for red wine consumption or white wine consumption [standardized rate ratio (SRR) 0.98, 95% CI 0.68 to 1.40 and SRR 0.95, 95% CI 0.69 to 1.32, respectively]. When evaluating the amount of intake, light to moderate drinkers (<2 drinks per day) showed a lower risk (SRR 0.93, 95% CI 0.80 to 1.08) than heavy drinkers (≥2 drinks per day) (SRR 1.00, 95% CI 0.86 to 1.16). Overall, no association could be found between wine consumption and the risk of colorectal cancer [121].

Finally, a meta-analysis by Schwingshackl et al. was aimed at evaluating the association between a MedDiet and the risk of cancer in 83 prospective studies. Among other components of the MedDiet, the authors found an inverse association between moderate alcohol intake and the risk of developing cancer (RR 0.89, 95% CI 0.85 to 0.93), when compared to higher intakes. The authors of this study stated that even though red wine contains beneficial compounds that may play a role in cancer, to date it is impossible to certify the effects of alcohol on tumor pathogenesis, and therefore it should not be recommended to abstinent individuals [34].

When evaluating the risk of cancer, it appears that dose and type of beverage are two main factors that affect this risk. Alcohol by itself, at high doses, seems to increase the risk of cancer, and the same seems to happen with wine intake. A recent review showed that alcohol intake at low doses does not increase the risk of cancer, with the exception of breast and prostate cancer [135]. However, red wine seems to have a more protective effect when compared to other alcoholic beverages.

### 2.5. Dementia

Dementia is a progressive neurodegenerative disorder and is a major cause of disability worldwide. Modifiable lifestyle factors, including diet and specific foods, have been addressed as risk factors for this disease with inconclusive results [136,137]. A meta-analysis published by Xu et al. in 2017 evaluated the association between alcohol intake and dementia in order to better define the quantity of alcohol that could significantly increase the risk of this disease. A U-shaped association was observed between alcohol consumption and the risk of dementia, in which light, light-to-moderate and moderate intake showed an inverse association, while heavier doses showed higher risk, even though there was no statistical significance. Moreover, only wine showed a trend towards a protective effect for dementia when comparing current drinkers with never drinkers (RR 0.67, 95% CI 0.48 to 0.94; *p* = 0.2) or light-to-moderate drinker versus none (RR0.58, 95% CI 0.39 to 0.87; *p* = 0.196), among other types of alcoholic beverages. Furthermore, the authors identified the dose of alcohol intake to be associated with the lowest risk of dementia, revealing that intake ≤ 12.5 g/day decreased the risk. The authors concluded that a modest quantity of alcohol consumption might help decrease the risk of dementia, mainly due to its higher polyphenolic content [122].

## 3. Conclusions

The information regarding the effects of wine on different health outcomes has not varied widely over the last five years. Inconclusive and contradictory studies remain an important part of the available information, mainly because of the different populations, alcohol dosage, and study design used in the analyses. For now, light to moderate wine intake seems to have some beneficial effects on NCD, such as hypertension, cancer, dyslipidemia and dementia. We can agree that high doses of any alcohol intake, including wine, are harmful and should be avoided. Nonetheless, there is still a long way to go before definitive recommendations on wine intake can be made.

## Figures and Tables

**Table 1 molecules-25-05045-t001:** Summary of study characteristics, outcomes and main results of studies included in the analysis.

Reference	Design, Subjects (n), Follow-up	Population	Intervention/Dose	Outcomes	Main Results
Y. Gepner et al. [104]	Randomized controlled trial, n = 54, 6 months	Adults, T2D, alcohol abstainers	150 mL water, white wine, or red wine	BP (24-h ABPM)	Moderate daily red wine intake (150 mL) had no effect on mean daily BP, but showed transient hypotensive response at midnight (3–4 h after ingestion), decreasing SBP −10.6 mmHg (95% CI −14.1 to −0.6; *p* = 0.03) and DBP −7.7 mmHg (−11.8 to 0.9; *p* = 0.076).
T.A. Mori et al. [105]	Randomized controlled trial, cross-over design, n = 28, 16 weeks	Adults, T2DM, men and post-menopausal women, regular drinkers	Red wine or DRW 230 mL/day for women and 300 mL/d for men, or water.	Effect of wine consumption on 24 h ambulatory BP, heart rate and other markers	Red wine significantly increased awake SBP (2.5 ± 1.2 mmHg; *p* = 0.033) and DBP (1.9 ± 0.7 mmHg; *p* = 0.008) compared to water and decreased DBP overnight (2.0 ± 0.8 mmHg; *p* = 0.016) compared to DRW. Nonetheless, there was no significant overall effect of red wine on mean 24 h SBP or DBP. Red wine had no effect on TC, TG, HDL-C, LDL-C, fasting glucose and insulin levels, or HOMA-IR score.
S. McDonagh et al. [106]	Randomized, crossover trial, n = 12, 2 weeks	Healthy normotensive men	175 mL red wine, vodka or water	BP response to NO_3_^−^ rich salad and red wine	Red wine and NO_3_^−^ rich salad lowered SBP at 2 h (−5 mmHg) and 5 h (−4 mmHg) and DBP (2–4 mmHg) after intake.
I. Roth et al. [107]	Randomized controlled trial, cross-over design, n = 38, 10 weeks	Adults, men, T2DM or ≥3 cardiovascular risk factors	30g ethanol from white wine or gin	Effect of white wine on BP and plasma NO concentration	White wine decreased SBP (−4.91 mmHg, 95% CI −9.41 to −0.42; *p* = 0.033) and DBP (−2.90, 95% CI −5.50 to −0.29; *p* = 0.030) significantly compared to gin (*p* < 0.040); and significantly increased plasma NO concentrations (27.86, 95% CI −6.86 to 62.59; *p* = 0.013).
M.T. García-Conesa et al. [108]	Meta-analysis of 128 human randomized controlled trials (n = 5538)	Adults, distributed over five continents	250 to 400 mL red wine	Association between intake of wine and other foods on different biomarkers of cardio-metabolic risk	Anthocyanin rich products (wine/red grapes) reduced systolic (−3.31 mmHg; *p* = 0.014) and diastolic (−1.50 mmHg; *p* = 0.002) BP, but increased Hb1Ac (+0.26; *p* = 0.026)
S. Weaver et al. [109]	Meta-analysis, 37 studies	Adults, healthy or T2DM/obesity/MS	RWP supplementation (dose ND)	Effect of RWP on vascular health	RWP significantly improved SBP (−2.6 mmHg, 95% CI −4.8 to −0.4; *p* = 0.010), especially in at risk population (−3.2 mmHg, 95% CI −5.7 to −0.8; *p* = 0.010)
J. Ye et al. [110]	Meta-analysis, 9 studies, N/D	Adults, T2DM	Red wine 120–360 mL/d	Effect of wine intake on BP, glucose parameters and lipid profile in T2DM	Red wine intake significantly reduced DBP (MD 0.10, 95% CI 0.01–0.20; *p* = 0.03). No significant differences in glucose or lipid parameters.
Y. Gepner et al. [111]	Randomized controlled trial, n = 224, 2 years	Adults, 40–75 years with T2DM	150 mL of red wine or white wine	Changes in lipid profile (HDL-C, apolipoprotein (a)_1_, TC/HDL-C ratio) and glycemic control (FPG, HOMA-IR)	Red wine intake increased HDL-C (2.9 mg/dL, 95% CI 1.6–2.2 mg/dL; *p* < 0.001) and apolipoprotein (a)_1_ (0.03 g/L, 95% CI 0–0.06 g/L; *p* = 0.05), and decreased TC/HDL-C ratios (0.27, 95% CI −0.52 to −0.01; *p* = 0.039). White wine decreased FPG (−17.2 mg/dL, 95% CI −28.9 to −5.5 mg/dL; *p* = 0.004) and HOMA-IR score (−1.2, 95% CI −2.1 to −0.2; *p* = 0.019)
K. Abraham et al. [112]	Randomized controlled trial, n = 9, 2 weeks	Adults, T2DM and pre-diabetic	263 mL red wine or water	Acute effect of red wine in glycemic control	Greater insulin iAUC response after wine intake (50%; *p* < 0.05), but no change in glucose iAUC (*p* = 0.82)
J. Huang et al. [113]	Meta-analysis, 13 prospective studies, 397,296 subjects	Adults, T2DM or healthy	Stratified in 0–10 g/day, 10–20 g/day or >20 g/day	Risk of T2DM	Wine intake was associated with 15% reduction in T2DM risk (RR 0.85, 95% CI 0.80–0.89), with a peak risk reduction at 20–30g/d
J. Woerdeman et al. [114]	Randomized controlled trial, n = 30, 8 weeks	Adults, obese (BMI ≥30 kg/m^2^), white ethnicity, healthy	RWP extract 600 mg/d or placebo	Effect of supplementation of RWP on insulin sensitivity in obese adults	RWP supplementation did not alter insulin sensitivity nor lipid profile compared to placebo (M-value (mg/kg/min) 3.3, CI 2.4–4.8 vs. 2.9, CI 2.8–5.9; *p* = 0.65, respectively)
R. Golan et al. [115]	Randomized controlled trial, n = 224, 2 years	Adults, T2DM, abstainers	16.9 g of ethanol from dry red wine (150 mL), or 15.8 g from white wine (150 mL)	Effect of moderate wine intake in atherosclerosis	Moderate wine intake was associated with no progression in carotid total plaque volume (−1.2 mm^3^, SD 16.9, CI −3.8 to 6.2; *p* = 0.6 for white wine; −1.3, mm^3^, SD 17.6, CI −3.4 to 6.0; *p* = 0.5 for red wine) and with a small regression among those with higher carotid plaque burden at baseline (mean −0.11; *p* = 0.04)
M. Taborsky et al. [116]	Randomized controlled trial, n 157, 12 months	Adults, healthy, mild to moderate cardiovascular risk	Red or white wine, 0.2 L/day in women <70 kg and 0.3 L/d in women <70 kg and men	Effect of regular red and wine intake in HDL-C and other markers of atherosclerosis	HDL-C significantly decreased at 6 months in the white wine group (−0.14 (SD 0.41); *p* = 0.005), no changes for red wine. LDL-C significantly decreased in both groups at 6 months (−0.39 (0.74); *p* < 0.001 for white wine and −0.27 (0.68); *p* < 0.001 for red wine) and at 12 months (−0.24 (0.73); *p* = 0.003 for white wine and −0.24 (0.78); *p* = 0.013 for red wine) compared with baseline. A significant reduction in TC was observed at 6 months in both groups (−0.32 (1.13); *p* = 0.017 for white wine and −0.33 (0.82); *p* = 0.001 for red wine), but only for red wine at 12 months (−0.24 (0.82); *p* = 0.016).
L. di Renzo et al. [117]	Randomized controlled trial, n 55, 1 day	Healthy adults	30 g of ethanol from red wine, white wine or vodka	Effect of ethanol and polyphenols present in alcoholic beverages on oxidative status when eating an antioxidant meal	Red wine intake during a HFM significantly reduced Ox-LDL-C levels (−4.97 ± 33.18; *p* < 0.05) compared with HFM alone. Red wine significantly up-regulated CAT gene expression (fold change 4.04)
Fang et al. [118]	Meta-analysis, 76 observational studies, n = 6,316,385 subjects, 11.4 years (3.3–30y)	Adults, general population	Dose ND	Association between gastric cancer and dietary factors	Alcohol consumption increased gastric cancer risk (RR 1.15, 95% CI 1.01–1.31), nonetheless wine did not significantly increase this risk (RR 1.02, 95% CI 0.77–1.34).
J.Y. Chen et al. [119]	Meta-analysis, 26 observational studies, n = 18,106 subjects	Adult women with breast cancer	1 drink or 12.5 g of ethanol	Association between wine dose and breast cancer risk	Wine intake increased breast cancer risk (RR 1.36; 95% CI 1.20–1.54; *p* < 0.001), with a dose-response association, showing a 0.59% increase for each increment of 1g/day of ethanol from wine. However, risk decreased in women consuming <80g/day of wine (10g ethanol), with lowest risk at 40g/day of wine (5g/day ethanol).
M.D. Vartolomei et al. [120]	Meta-analysis, 174 studies, n = 455,413 subjects	Adults, overall population	Moderate red wine intake (ND)	Effect of red wine on prostate cancer development	Moderate red wine consumption was associated with lower risk of prostate cancer (RR 0.88, 95% CI 0.78–0.999; *p* = 0.047)
W. Xu et al. [121]	Meta-analysis, 17 observational studies, n = 12,110 subjects	Adults, general population	Stratified in non-drinkers plus occasional drinkers (<0.5 drinks/day), light to moderate drinker (<2 drinks/day) and heavy drinkers (≥2 drinks/day)	Effects of wine intake on colorectal cancer risk	Any wine consumption did not affect colorectal cancer risk versus nondrinkers (RR 0.99, 95% CI 0.89–1.10). No difference among men and women (0.88, CI 0.66–1.18 and 0.83, CI 0.67–1.03, respectively), red or white wine (0.98, CI 0.68–1.40, and 0.95, CI 0.69–1.32, respectively) nor drinking category (light to moderate 0.93, CI 0.80–1.08, and heavy drinking 1.00, CI 0.86–1.16).
L. Schwingshackl et al. [34]	Meta-analysis, 83 prospective studies, n = 2,130,753 subjects	Adults, overall population	Moderate red wine intake in a Mediterranean diet (ND)	Cancer risk and cancer mortality risk	Inverse association for moderate alcohol intake and cancer risk (RR 0.89, 95% CI 0.85–0.93)
W. Xu et al. [122]	Meta-analysis, 16 observational studies, 3–25 years	Adults, general population	Stratified in light (<7 drinks/week), light-to-moderate (< 14drinks/ week), moderate (7–14 drinks/ week) moderate-to-heavy (>7 drinks/week) and heavy drinkers (>14 drinks/week)	Association between quantity of alcohol intake and risk of dementia	U-shaped association between alcohol consumption and risk of dementia, Wine showed a trend towards a protective effect for dementia, for current drinkers versus never drinkers (RR 0.67, 95% CI 0.48–0.94; *p* = 0.2) or light-to-moderate drinker versus non-drinkers (RR 0.58, 95% CI 0.39–0.87; *p* = 0.196).

24-h ABPM: ambulatory blood pressure measurement; BMI: body mass index; BP: blood pressure; CAT: catalase; CI: confidence interval; DBP: diastolic blood pressure; DRW: dealcoholized red wine; FPG: fasting plasma glucose; Hb1Ac; glycated hemoglobin; HDL-C: high-density lipoprotein cholesterol; HFM: high-fat meal; HOMA-IR: Homeostatic Model Assessment for Insulin Resistance; iAUC: incremental blood glucose area under the curve; LDL-C: low-density lipoprotein cholesterol; MS: metabolic syndrome; ND: non-defined; NO: nitric oxide; NO^3−^: nitrate; Ox-LDL-C: Oxidized low-density lipoprotein cholesterol; RR: relative ratio; RWP: Red wine polyphenols; SPB: systolic blood pressure; T2DM: type 2 diabetes mellitus; TC: total cholesterol; TG: tryglicerides.

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
