# Peer review of "Wine Intake in the Framework of a Mediterranean Diet and Chronic Non-Communicable Diseases: A Short Literature Review of the Last 5 Years"

_molecules, 2020, doi:10.3390/molecules25215045_

Round 1
Reviewer 1 Report
Overall this is a well-written study and an interesting review. Please see my suggestions for revision below:
Please justify why you have selected to include only the last 5 years in your literature review.
Paragraph starting on line 62: In this paragraph, it would be of benefit to mention some of the findings regarding increased alcohol intake and hypertension:
Roerecke M, et al. Sex-specific associations between alcohol consumption and incidence of hypertension: a systematic review and meta-analysis of cohort studies. J Am Heart Assoc 2018;7:e008202.
Wood AM, et al. Risk thresholds for alcohol consumption: combined analysis of individual-participant data for 599 912 current drinkers in 83 prospective studies. Lancet 2018;391:1513-23.
Roerecke M, Kaczorowski J, Tobe SW, et al. The effect of a reduction in alcohol consumption on blood pressure: a systematic review and metaanalysis. Lancet Public Health 2017;2:e108-20.
Line 100: delete the question marks “??” after the word “formulations”
Line 108: I suggest replacing “gender” (which describes behavior) with “sex” (which describes the biological aspect).
Line 121: Please make sure the abbreviation “ANC” is defined in the manuscript.
Line 138: Please make sure the abbreviation “BP” is defined in the manuscript.
Paragraph starting on line 168: It appears you are conducting a systematic review; therefore, please also include what you considered the “comparator” in studies (i.e. no wine intake?) and the outcomes you were interested in.
Line 171: Please clarify what you mean by “articles published in relevant impact factor journals”
Line 171: Please indicate the number of studies you excluded because they were not in English and Spanish. This is a potential source of bias in your review.
Please provide more detail on your systematic review methodology…i.e. how many reviewers were involved in each stage and whether you used an instrument to measure the quality of included studies (e.g. the Cochrane risk of bias tool or other instrument).
Please include a PRISMA flow diagram to summarize your literature search and study selection.
Line 202: Please ensure you have defined the abbreviation MDP in the manuscript.
Line 425: Delete the question marks “???” at the end of this sentence.
Author Response
Overall this is a well-written study and an interesting review. Please see my suggestions for revision below:
- Please justify why you have selected to include only the last 5 years in your literature review.
Comment 1: Thank you for your comments. We decided to review the most relevant articles published only during the past 5 years due to two main reasons. In first place, there is a big amount of articles related to this topic, so we decided that, by evaluating a fewer amount of years, the quantity of articles needed to be reviewed would be less, making it possible to asses each one in a more thorough way. On the other hand, the information published about this topic has not varied widely during the past 5 or 10 years, so reviewing 10 or 5 years would not make such a difference in the results presented.
- Paragraph starting on line 62: In this paragraph, it would be of benefit to mention some of the findings regarding increased alcohol intake and hypertension:
Comment 2: Thank you for your comments. We have added the information requested. The changes are highlighted in yellow (Lines 71-80).
- Line 100: delete the question marks “??” after the word “formulations”
Comment 3: The questions marks have been deleted.
- Line 108: I suggest replacing “gender” (which describes behavior) with “sex” (which describes the biological aspect).
Comment 4: The word “gender” has been replaced by the word “sex”. The change is highlighted in yellow (line 121).
- Line 121: Please make sure the abbreviation “ANC” is defined in the manuscript.
Comment 5: This abbreviation was previously defined in line 96.
- Line 138: Please make sure the abbreviation “BP” is defined in the manuscript.
Comment 6: This abbreviation was previously defined in line 45.
- Paragraph starting on line 168: It appears you are conducting a systematic review; therefore, please also include what you considered the “comparator” in studies (i.e. no wine intake?) and the outcomes you were interested in.
Comment 7: Thank you for your comments. We have not conducted a systematic review. Instead, the objective of this manuscript was to update the available information about wine intake and its relationship with chronic diseases. We only reviewed the available literature in the context of a comprehensive review; therefore, we delete the methodology section so there would be no confusions. The changes are highlighted in yellow (lines 227-229).
- Line 171: Please clarify what you mean by “articles published in relevant impact factor journals”
Comment 8: We decided to include studies published in important journals, qualified in the highest quartiles of publications. Nonetheless, we decided to eliminate this paragraph so there would be no misunderstandings as to whether this review is or isn’t systematic.
- Line 171: Please indicate the number of studies you excluded because they were not in English and Spanish. This is a potential source of bias in your review.
Comment 9: Again, we explain this is not a systematic review and the methods have been changed so it would not cause confusion.
- Please provide more detail on your systematic review methodology…i.e. how many reviewers were involved in each stage and whether you used an instrument to measure the quality of included studies (e.g. the Cochrane risk of bias tool or other instrument).
Comment 10: Since this is not a systematic review, we have changed the methods to avoid misunderstandings.
- Please include a PRISMA flow diagram to summarize your literature search and study selection.
Comment 11: Since this is not a systematic review, we have changed the methods to avoid misunderstandings.
- Line 202: Please ensure you have defined the abbreviation MDP in the manuscript.
Comment 12: This abbreviation was previously defined in line 48.
- Line 425: Delete the question marks “???” at the end of this sentence.
Comment 13: The questions marks have been deleted.
Reviewer 2 Report
The authors present a work entitled ".Wine intake and chronic non-communicable diseases: 3 short literature review of the last 5 years". The review was organized considering the work of the last 5 years, so it only reports the most recent novelties of the literature. The work setting is very good and the manuscript looks very fluid and easy to read. The topic is of great interest to readers and the scientific world. In order to further improve the quality of the manuscript, I suggest only a few changes:
Pag 2 Ln 61: this is a very interesting point, please, increase the number of references in this regard
The authors mention the smell of wine but never pause to sufficiently discuss the volatile fraction (terpenes, aldehydes ... VOCs) to which fragrance and relevant health actions are ascribable. Could the authors enrich this aspect?
Pag 3 Ln 77-79: this statement is a crucial point, please enlarge the refernces in this regard
Please, correct the formatting of reference 25
Overall, wine is only one of the components of the Mediterranean diet, and is considered in this work. However, in a not always fluid way, the Mediterranean diet is mentioned several times while the speech should address the intake of wine as a priority.
Section 1.2 Wine polyphenols in human health: in my opinion, this is the core of the article, please, enrich this section with references and your comments.
Author Response
The authors present a work entitled ".Wine intake and chronic non-communicable diseases: 3 short literature review of the last 5 years". The review was organized considering the work of the last 5 years, so it only reports the most recent novelties of the literature. The work setting is very good and the manuscript looks very fluid and easy to read. The topic is of great interest to readers and the scientific world. In order to further improve the quality of the manuscript, I suggest only a few changes:
- Pag 2 Ln 61: this is a very interesting point, please, increase the number of references in this regard
Comment 1: Thank you for your comments. The number of references addressing this point has been increased. The changes are highlighted in yellow (Line 61; Reference section, references 16-22).
- The authors mention the smell of wine but never pause to sufficiently discuss the volatile fraction (terpenes, aldehydes ... VOCs) to which fragrance and relevant health actions are ascribable. Could the authors enrich this aspect?
Comment 2: Thank you for this comment. A brief paragraph addressing this issue has been added. The changes are highlighted in yellow (Lines 135-160).
- Pag 3 Ln 77-79: this statement is a crucial point, please enlarge the references in this regard
Comment 3: Thank you for your comment. The references have been reviewed and increased to support this point. The change is highlighted in yellow (Line 91).
- Please, correct the formatting of reference 25
Comment 4: The formatting of reference 25 has been reviewed.
- Overall, wine is only one of the components of the Mediterranean diet, and is considered in this work. However, in a not always fluid way, the Mediterranean diet is mentioned several times while the speech should address the intake of wine as a priority.
Comment 5: Thank you for your comment. We have slightly changed the title of the review and the objectives, as we would like to address the benefits of wine intake in the context of a Mediterranean diet. The changes are highlighted in yellow (Lines 2-4).
- Section 1.2 Wine polyphenols in human health: in my opinion, this is the core of the article, please, enrich this section with references and your comments.
Comment 6: Thank you for this comment. A brief paragraph addressing this issue has been added. The changes are highlighted in yellow (Lines 128-132; 133-134; 135-160; 170-176; 188-194; 195; 207; 214;222).
Round 2
Reviewer 1 Report
Lines 71 to 80: I think you have misinterpreted the studies in reference 28 and reference 30. For reference 28, you indicate a reduction in blood pressure with alcohol intake, but I think the opposite was true (i.e. an increase in blood pressure with alcohol intake).
For reference 30 you state that those drinking 100g alcohol per week had lower all-cause mortality. This is a misinterpretation of the article, which showed those drinking any amount greater than this had higher mortality risk.
Line 154: Delete "in"
Line 172: Replace "related" with "associated"
Line 218: Please re-word this sentence
Author Response
Lines 71 to 80: I think you have misinterpreted the studies in reference 28 and reference 30. For reference 28, you indicate a reduction in blood pressure with alcohol intake, but I think the opposite was true (i.e. an increase in blood pressure with alcohol intake).
For reference 30 you state that those drinking 100g alcohol per week had lower all-cause mortality. This is a misinterpretation of the article, which showed those drinking any amount greater than this had higher mortality risk.
Comment 1: Thank you to Reviewer for your suggestion. We have rewritten this paragraph. The changes are highlighted in yellow (lines 71-75; 79-82)
Line 154: Delete "in"
Comment 2: It was deleted.
Line 172: Replace "related" with "associated"
Comment 3: It was associated.
Line 218: Please re-word this sentence
Comment 4: It was rewritten. The changes are highlighted in yellow (lines 218-224)